# Effectiveness of Combinational Treatments for Alzheimer’s Disease with Human Neural Stem Cells and Microglial Cells Over-Expressing Functional Genes

**DOI:** 10.3390/ijms24119561

**Published:** 2023-05-31

**Authors:** Young-Hwan Ban, Dongsun Park, Ehn-Kyoung Choi, Tae Myoung Kim, Seong Soo Joo, Yun-Bae Kim

**Affiliations:** 1College of Veterinary Medicine, Chungbuk National University, Cheongju 28644, Chungbuk, Republic of Korea; 2Department of Biology Education, Korea National University of Education, Cheongju 28173, Chungbuk, Republic of Korea; 3Central Research Institute, Designed Cells Co., Ltd., Cheongju 28576, Chungbuk, Republic of Korea; 4College of Life Science, Gangneung-Wonju National University, Gangneung 25457, Gangwon, Republic of Korea

**Keywords:** Alzheimer’s disease, cognitive function, neural stem cell, microglial cell, functional gene, choline acetyltransferase, neprilysin, scavenger receptor A, acetylcholine, amyloid-β peptide

## Abstract

Alzheimer’s disease (AD) is one of the most common neurodegenerative diseases. In AD patients, amyloid-β (Aβ) peptide-mediated degeneration of the cholinergic system utilizing acetylcholine (ACh) for memory acquisition is observed. Since AD therapy using acetylcholinesterase (AChE) inhibitors are only palliative for memory deficits without reversing disease progress, there is a need for effective therapies, and cell-based therapeutic approaches should fulfil this requirement. We established F3.ChAT human neural stem cells (NSCs) encoding the choline acetyltransferase (ChAT) gene, an ACh-synthesizing enzyme, HMO6.NEP human microglial cells encoding the neprilysin (NEP) gene, an Aβ-degrading enzyme, and HMO6.SRA cells encoding the scavenger receptor A (SRA) gene, an Aβ-uptaking receptor. For the efficacy evaluation of the cells, first, we established an appropriate animal model based on Aβ accumulation and cognitive dysfunction. Among various AD models, intracerebroventricular (ICV) injection of ethylcholine mustard azirinium ion (AF64A) induced the most severe Aβ accumulation and memory dysfunction. Established NSCs and HMO6 cells were transplanted ICV to mice showing memory loss induced by AF64A challenge, and brain Aβ accumulation, ACh concentration and cognitive function were analyzed. All the transplanted F3.ChAT, HMO6.NEP and HMO6.SRA cells were found to survive up to 4 weeks in the mouse brain and expressed their functional genes. Combinational treatment with the NSCs (F3.ChAT) and microglial cells encoding each functional gene (HMO6.NEP or HMO6.SRA) synergistically restored the learning and memory function of AF64A-challenged mice by eliminating Aβ deposits and recovering ACh level. The cells also attenuated inflammatory astrocytic (glial fibrillary acidic protein) response by reducing Aβ accumulation. Taken together, it is expected that NSCs and microglial cells over-expressing ChAT, NEP or SRA genes could be strategies for replacement cell therapy of AD.

## 1. Introduction

Alzheimer’s disease (AD) is involved in 70% of dementia cases worldwide in elderly people [1]. It leads to the loss of mental processing ability, including communication, thinking, judgment and physical abilities. AD also imposes enormous emotional and financial burdens through the provision of care [2].

In AD patients, cholinergic dysfunction is one of the primary causes of cognitive disorders, in which decreased activity of the enzyme choline acetyltransferase (ChAT) responsible for acetylcholine (ACh) synthesis in cholinergic neurons is observed [3,4]. AD is also characterized by the accumulation of amyloid-β (Aβ) peptides in the brain [5]. Although the cause of AD is uncertain, increased production and deposition of Aβ has been the central conceptual framework for AD [6,7]. Aβ is generated from Aβ precursor proteins (APP) through sequential cleavages first by β-secretase and then by γ-secretase complex. In normal brains, the physiological concentration of the Aβ peptides has been indicated to be involved in modulating neurogenesis and synaptic plasticity. However, excessive Aβ production, aggregation, and deposition deleteriously affect many biologically-important pathways, leading to neural cell death [8].

To date, one of the AD therapies is largely based on using acetylcholinesterase (AChE) inhibitors designed to increase ACh concentration in the brain [3,9]. Other therapeutics available are *N*-methyl-d-aspartate (NMDA) receptor antagonists [1,2]. Both therapies contribute minimal impact on the disease and decelerate the progression of the disease, providing symptomatic relief, but fail to achieve a definite cure, since the therapies using AChE inhibitors and NMDA receptor antagonists are only palliative. Among diverse efforts being directed towards several targets of AD, now cell therapy should be needed to recover ACh concentration in the brain and eliminate causative Aβ peptides over the long term, thus, lately receiving considerable attention as a potential therapeutic option [10].

We have previously demonstrated that transplantation of HB1.F3 (shortly F3), a human neural stem cell (NSC) line, encoding diverse functional genes preserved host neurons and recovered physical functions in animal models of Parkinson’s disease (PD), cerebral palsy (CP) and stroke [11,12,13]. For AD therapy, we established F3.ChAT cells over-expressing the human ChAT gene by transfecting the ChAT gene into the HB1.F3 cell line—the F3.ChAT cells recovered brain ACh level and memory function in aged mice and transgenic and neurotoxin-induced AD model animals [4,14,15,16].

On the other hand, increased uptake and degradation of toxic Aβ molecules have been a significant research and therapeutic target [17,18]. Microglial cells, which are the resident macrophages in the central nervous system (CNS) and migrate toward sites of Aβ accumulation, play a critical role in clearing Aβ through phagocytic activities mediated by scavenger receptor A (SRA) and receptors for advanced glycosylation end-products (RAGE) [7,19]. Neprilysin (NEP) is a member of the metalloprotease 13 family of zinc metalloproteases [20]. It has been demonstrated that NEP could cleave Aβ peptides in vitro and in vivo [20,21,22]. Indeed, combinational administration of Aβ and thiorphan, an inhibitor of NEP, resulted in a significant intracellular accumulation of Aβ in the neurons, accompanied by neuronal atrophy and loss in monkeys [23]. Moreover, immunohistochemistry revealed degeneration of ChAT-positive cholinergic neurons and an increase of glial fibrillary acidic protein (GFAP)-positive astrocytes. Therefore, NEP is considered one of the most essential enzymes for controlling Aβ levels [20,21].

HMO6, a human microglial cell line, was first established earlier, showing expression profiles of cytokines and chemokines with the isolated primary [24]. For AD therapy, we established HMO6.NEP cells over-expressing NEP, an Aβ-degrading enzyme, by transfecting the human NEP gene to the HMO6 human microglial cell line. Separately, macrophage SRA is the main receptor responsible for Aβ clearance [25]. It was reported that SRA-mediated interaction with Aβ promoted Aβ phagocytosis and clearance and down-regulated inflammatory genes [26]. Thus, HMO6.SRA cells over-expressing SRA, an Aβ-uptaking receptor, were also established by transfecting the human SRA gene to HMO6 cells.

Several experimental animal models of AD have been developed. The animal models include cholinergic dysfunction-, Aβ peptide- and neuroinflammation-related memory impairments [27,28]. AF64A, a specific cholinotoxin, was found to decrease the release of ACh and thereby induce cognitive impairments, including learning and memory deficits as a salient feature of AD [4]. Ample experimental evidence indicates that intracerebroventricular (ICV) injection or infusion of Aβ into rodents induces learning and memory deficits and neurodegeneration in the brain areas related to cognitive function [29]. The consequence of elevated Aβ by lipopolysaccharide (LPS) injection could cause neuronal cell death, and this may be associated with memory impairments [30,31]. However, the cognitive deficits last for a short period after challenging with Aβ or LPS, possibly due to the early elimination of the toxins and anti-inflammatory process in the brain [32].

In the present study, we established a proper AD model showing a high-level accumulation of Aβ and long-term cognitive deficits from various experimental animal models. To the model animals, we transplanted F3.ChAT NSCs, HMO6.NEP and HMO6.SRA microglial cells each or their combinations (F3.ChAT + HMO6.NEP or F3.ChAT + HMO6.SRA) to identify whether these cell therapies alleviate Aβ burden, increase ACh levels and restore cognitive function.

## 2. Results

### 2.1. Establishment of NSCs and Microglial Cells Over-Expressing Functional Genes

In the established human NSCs and microglial cells encoding functional genes, the expressions of ChAT, NEP and SRA mRNAs were confirmed to be higher in F3.ChAT, HMO6.NEP and HMO6.SRA cells, respectively, than those in each matching parental cell (Figure 1A & Appendix A). In parallel with the mRNA expression, immunocytochemical staining to detect functional protein production revealed ChAT immunoreactivity stronger in F3.ChAT cells than in parental F3 cells (Figure 1B). Enhanced production of NEP and SRA proteins in HMO6.NEP and HMO6.SRA cells, respectively, more than in HMO6 cells, were also observed—notably, HMO6.NEP cells eliminated Aβ from the culture medium in a time-dependent manner (Figure 1C)—more early and remarkable Aβ clearances by HMO6.SRA cells were confirmed, in comparison with a negligible effect by HMO6 cells.

### 2.2. Establishment of AD Animal Models

#### 2.2.1. Impairment of Cognitive Function by Neurotoxicants

The mice in all groups challenged ICV or intraperitoneally (IP), with each neurotoxicant, displayed severe impairment of learning and memory functions as measured by passive avoidance (Figure 2A). Four weeks after injection, however, the memory impairments were maintained only in AF64A(ICV), AF64A(IP) and Aβ(ICV) + HFD groups as observed by both the passive avoidance (Figure 2B) and Morris’s water-maze performances (Figure 2C). By comparison, LPS(IP) and Aβ(ICV) + ND group mice near-fully recovered their cognitive function, and LPS(ICV) exhibited partial recovery.

#### 2.2.2. Changes in Brain Aβ and ACh

Four weeks after toxicant injection, brain Aβ levels markedly increased in Aβ(ICV) + HFD, AF64A(ICV), AF64A(IP) and LPS(ICV) group animals (Figure 3A), in which AF64A(ICV) was the most effective. By comparison, Aβ(ICV) + ND and LPS(IP) was ineffective for Aβ accumulation (Appendix A). In an inverse relationship, Aβ(ICV) + HFD, AF64A(ICV) and AF64A(IP) challenges significantly decreased ACh concentrations in the brain tissues to 50% of normal mice (Figure 3B). LPS(ICV) also lowered ACh concentration to about 70% of the control value, while Aβ(ICV) + ND and LPS(IP) injection did not affect the ACh level.

#### 2.2.3. Selection of an AD Animal Model

A good relationship among Aβ accumulation, ACh depletion and cognitive dysfunction was achieved. Such results indicate that AF64A could be a proper animal model displaying long-term learning and memory impairments mediated by Aβ accumulation and ACh depletion for cell therapy assessment. So, we adopted AF64A(ICV) model for the present study.

### 2.3. Evaluation of Therapeutic Efficacies

#### 2.3.1. Improvement of Cognitive Function

The therapeutic effects of stem cells and microglial cells encoding functional genes were investigated in AF64A-challenged AD model animals. Three days after AF64A injection, the mice displayed profound impairments of memory acquisition and retention in passive avoidance (Figure 4A) and Morris’s water-maze (Figure 4B) performances.

Such cognitive dysfunction in mice challenged with AF64A lasted four weeks (Figure 4C). However, the memory deficits were markedly ameliorated by treatment with F3.ChAT, F3.ChAT + HMO6.NEP or F3.ChAT + HMO6.SRA cells, in which F3.ChAT + HMO6.SRA was the most effective. HMO6.SRA alone was somewhat effective in passive avoidance performance but not in Morris’s water-maze test, while the efficacies of F3, HMO6, and HMO6.NEP were relatively low.

#### 2.3.2. Expression of Functional Genes in Transplanted Cells

Four weeks after ICV transplantation, the transplanted human NSCs (F3.ChAT) and microglial cells (HMO6.NEP and HMO6.SRA) survived in the mouse brain as observed by immune-positive reaction to human mitochondria (hMito) (Figure 5). The cells were detected in injury sites, including periventricular regions, hippocampus, thalamus, hypothalamus, and cortices. By double immunostaining, it was confirmed that the human cells clearly showed production of each functional protein, that is, ChAT, NEP, or SRA for F3.ChAT, HMO6.NEP and HMO6.SRA, respectively, is indicative of long-term survival and expression of functional genes.

#### 2.3.3. Aβ Elimination and ACh Recovery

In ELISA, four weeks after the AF64A challenge, Aβ level in the brain significantly increased (Figure 6A). However, the AF64A-induced accumulation of Aβ peptides was reversed by treatment with F3.ChAT, HMO6.NEP, HMO6.SRA, and more effectively by their combinations.

In parallel with the decrease in Aβ level, ACh concentration in the brain challenged with AF64A markedly decreased to a half level (Figure 6B). The ACh was significantly recovered by transplantation of F3.ChAT cells, whereas F3 and HMO6 cells were ineffective. Although HMO6.NEP and HMO6.SRA cells alone displayed minimal effects, they further increased the effect of F3.ChAT treatment, in which HMO6.SRA was superior to HMO6.NEP cells.

#### 2.3.4. Aβ Elimination and Neuroprotection

In western blot analysis, the concentrations of Aβ and GFAP markedly increased to 3.55 and 2.55 folds, respectively, in the brain tissues of AF64A-challenged mice (Figure 7 & Appendix A). Although F3.ChAT, HMO6 and HMO6.NEP cells exhibited mild effects in lowering the Aβ level, they significantly decreased GFAP concentration. By comparison, HMO6.SRA alone remarkably reduced both the Aβ and GFAP levels. Notably, combination therapy of F3.ChAT + HMO6.NEP or F3.ChAT + HMO6.SRA further decreased the Aβ and GFAP than treated alone, wherein HMO6.SRA was more effective than HMO6.NEP cells.

## 3. Discussion

AD is one of the most devastating neurodegenerative diseases and burdens individuals and societies since about 10% of people older than 65 suffer from the disease [1]. Although the causes of AD are not well defined, the deposition of cytotoxic Aβ peptides and ensuing cholinergic neurotoxicity, depleting the ACh level, has been suggested to be the central conceptual framework for AD [4,7].

In many reports, direct injection of Aβ peptides into the mouse brain was found to be toxic to neurons and induced memory impairment [32,33]. However, the injected Aβ peptides rapidly disappeared, leading to the recovery of cognitive function in about two weeks. The present study also observed the full recovery of learning and memory functions four weeks post-injection of Aβ peptides in ND-fed mice. By comparison, Aβ injection to HFD-fed mice caused profound ACh depletion and cognitive dysfunction. The prolonged effect of Aβ on memory impairment in hyperlipidemic mice was also shown in our previous report [32]. Several studies suggested cholesterol is a risk factor for AD, which might be linked to Aβ production and deposition [34,35]. However, the Aβ deposition was relatively mild in four weeks, and the duration of cognitive dysfunction was not longer than six weeks in the present and previous studies [32]. Such short-term duration of memory deficits in Aβ-challenged mice may be due to the high activities of Aβ-degrading enzymes such as NEP and insulysin in the brain with intact microglial cells.

In contrast to the negligible effect of IP administration, ICV injection of LPS induced moderate Aβ accumulation, ACh depletion and cognitive dysfunction. Such inflammatory neurotoxicity and memory impairment were already demonstrated in previous studies [30,31,36]. However, the neural inflammation was transient and spontaneously recovered, displaying different features from the pathology of AD patients’ brains. Thus, the short-term and reversible models using Aβ peptide or LPS injections were not suitable for the efficacy evaluation of time-requiring cell therapy.

By comparison, ICV injection of AF64A induced long-term memory deficits following severe ACh depletion, parallel with profound Aβ accumulation, although its IP administration led to a relatively-low Aβ deposition. AF64A is a choline analogue taken up by the high-affinity choline transport system into cholinergic neurons and causes alterations in ChAT mRNA expression and enzyme activity [37,38]. Accordingly, it is well known that the cholinotoxin AF64A, inducing ACh depletion, has been used as an AD animal model [4]. Interestingly, in APPswe/PS1dE9 transgenic mice, it was demonstrated that functional changes of microglia, i.e., up-regulation of Aβ-producing systems (inflammatory cytokines and β-secretase) and down-regulation of Aβ-clearing machinery (scavenger receptor A and degrading enzymes such as NEP) played a key role for accumulation of neurotoxic Aβ peptides [39]. This is the reason why we adopted the AF64A model inducing the highest Aβ accumulation and profound long-term memory deficits for the evaluation of the efficacies of stem cells producing ACh (F3.ChAT) and microglial cells clearing Aβ (HMO6.NEP and HMO6.SRA).

Aβ production and clearance are key targets in developing disease-modifying therapeutic agents for AD [10,40]. Since microglial cells play significant roles in the internalization and degradation of Aβ peptides, the Aβ-degrading enzyme NEP in microglial cells has been considered a therapeutic target for AD [41,42]. A potential relationship between Aβ synthesis from the APP and NEP activity has been proposed [41]. On the other hand, SRA is the main receptor responsible for Aβ uptake and clearance [25]. SRA-mediated interactions with Aβ promote Aβ phagocytosis and clearance and down-regulate inflammatory genes [26]. Indeed, there were decreases in NEP and SRA contents followed by a high-level deposition of Aβ in the brain of mice challenged with AF64A in our preliminary findings. For these reasons, for cell therapy, we established F3.ChAT and HMO6.NEP, as well as HMO6.SRA cells focusing on ACh recovery and Aβ elimination, respectively.

F3.ChAT cells established exerted great efficacy in the treatment of AD mice challenged with AF64A. I.e., the cells successfully restored the brain ACh level by expressing the functional gene ChAT, thereby recovering memory function—such effects of F3.ChAT cells were also achieved in AD animal models, such as AF64A-induced cholinergic nerve injury and kainic acid-induced hippocampal injury rats [4,14], transgenic mice [16], as well as in aged mice [15]. In addition to the cognitive function-recovering effect, F3.ChAT cells reduced the Aβ and GFAP levels, suggesting anti-inflammatory and neuroprotective activities—such Aβ-eliminating effect of F3.ChAT cells were observed in our previous study [16]. It was also reported that human umbilical cord blood mesenchymal stem cells (UCBMSCs) reduced brain Aβ levels by up-regulating microglial NEP expression via the release of soluble intracellular adhesion molecule-1 (sICAM-1) [43]. Therefore, it is assumed that F3.ChAT NSCs also reduced Aβ levels by up-regulating the NEP of the host microglial cells. Furthermore, the elimination of the cytotoxic and inflammatory Aβ peptides may lower the astrocytic GFAP expression, a biomarker of brain injury [44], in addition to direct neuroprotective activities of growth factors (GFs) and neurotrophic factors (NFs) released from F3.ChAT cells [4,14,15,16,45].

HMO6.NEP and HMO6.SRA cells were also well established, as confirmed via western blotting and immunocytochemistry for functional proteins and Aβ-eliminating capacities in vitro. Functional change in microglial cells was observed in transgenic AD mice [39]. In this context, replacement therapy for the damaged and dead host microglia may be necessary to remove pre-existing and continuously-produced Aβ peptides in AD brains. HMO6.NEP and HMO6.SRA cells eliminated Aβ peptides in vitro by degrading and uptaking the molecules. Also, in vivo, treatment with the cells decreased Aβ levels. In addition, the cells decreased GFAP contents and recovered ACh concentration and cognitive function, suggestive of indirect and direct neuroprotective activities mediated by NFs and anti-inflammatory cytokines [46]. Such effects of HMO6.NEP and HMO6.SRA cells were obtained in Aβ-challenged mice, too [7].

Interestingly, in contrast to the inflammatory role of murine microglia aggravating AD, human microglia do not express inducible nitric oxide synthase (iNOS), producing NO, an important inflammatory mediator [47]. Instead, human microglial cells produce diverse GFs, NFs and Aβ-degrading enzymes exerting neuroprotective and Aβ- and tissue debris-clearing roles, showing different natures from murine microglia [46]. This is why human microglial cells could be a better tool for human CNS disease research than murine cells.

From the results of this study, the functional genes played vital roles: i.e., F3 and HMO6 parental cells did not exhibit considerable beneficial effects on ACh and cognitive functional recoveries, despite mild neuroprotective activity as observed in reduced GFAP contents. In contrast, the cells with functional genes were superior to their parental cells in all AD-therapeutic parameters. Furthermore, synergistic effects in Aβ elimination, ACh recovery, GFAP suppression, and cognitive functional restoration were achieved following the combinational treatments of F3.ChAT and HMO6.NEP or HMO6.SRA cells.

Notably, the transplanted NSCs and microglial cells distributed to injured brain sites, including periventricular, hippocampal, thalamic, and cortical regions, survived for longer than four weeks, and produced functional proteins (ChAT, NEP and SRA) by expressing transfected genes although the long-term survival up to 9 weeks and lesion-tropism of F3.ChAT cells were already confirmed in our previous reports [4,14,15]. More interestingly, the F3 NSCs survived for 24 months in the brain of non-human primates without immunosuppression [48] and did not cause tumor formation. Such findings suggest that human stem and microglial cells can survive for a long time in the human brain with no tumorigenic risk.

In summary, we demonstrated that combinational treatments of NSCs encoding the ChAT gene, an ACh-synthesizing enzyme, and microglial cells encoding the NEP gene, an Aβ-degrading enzyme, or SRA gene, an Aβ-uptaking receptor exerted complete restoration of learning and memory functions by eliminating Aβ accumulation and recovering ACh level in AF64A-challenged AD model mice. Interestingly, both important points, restoration of cholinergic nerve function and elimination of causative Aβ peptides to recover physical activity and stop additional neurodegeneration, were attained by the combinations of F3.ChAT and HMO6.NEP or HMO6.SRA cells. In conclusion, it is suggested that combinational therapeutic regimens could be candidates for the replacement therapy of AD as a long-term management strategy.

## 4. Materials and Methods

### 4.1. Establishment of NSCs and Microglial Cells

#### 4.1.1. Cell Culture

Human HB1.F3 NSC [49,50] and HMO6 microglial cell [24] lines from the University of British Columbia, BC, Canada (provided by Prof. SU Kim), were cultivated in Dulbecco’s Modified Eagle’s Medium (DMEM) supplemented with 10% fetal bovine serum (FBS), 100 U/mL penicillin and 100 μg/mL streptomycin (Invitrogen, Carlsbad, CA, USA). Cultures were maintained under 5% CO_2_ at 37 °C in tissue culture flasks. Media were changed every 2–3 days [7].

#### 4.1.2. Establishment of NSCs Encoding Functional Genes

An immortalized F3 NSC line was established from primary cultures of a 15-week gestational human fetal brain by infecting it with a retroviral vector encoding v-myc oncogene as described previously [49,50]. In addition, full-length human ChAT cDNA was obtained by a polymerase chain reaction (PCR) from human small intestine Marathon-Ready cDNA (Clontech, Mountain View, CA, USA) [51]. The human F3 NSC line was infected with a retrovirus encoding the human ChAT gene (plasmid pLPCX.ChAT), and puromycin-resistant colonies were selected [4,14]. The expression of ChAT mRNA was confirmed using a primer with sequences forward: 5′-CTGTGCCCCCTTCTAGAGC-3′; reverse: 5′-CAAGGTTGGTGTCCCTGG-3′.

#### 4.1.3. Establishment of Microglial Cells Encoding Functional Genes

HMO6.NEP and HMO6.SRA cells over-expressing human NEP and SRA were established by transfecting NEP or SRA, respectively, into human HMO6 microgial cell lines [7,24]. Full-length human NEP (Accession No. NM 000902) and SRA (Accession No. NM 138715) were obtained by PCR from the human small intestine using Marathon-Ready cDNA (Clontech). Plasmid DNA was sequenced with an ABI 3100 DNA sequencer (Applied Biosystems, Foster City, CA, USA). The confirmed plasmids were digested with EcoRI-XhoI and NheⅠ-HindⅢ, and the genes were inserted into the pcDNA3.1(+) mammalian expression vector. Then, pcDNA3.1(+) containing full-length cDNA was subcloned in *E. coli* (DH5α) and purified using a plasmid DNA miniprep kit (Qiagen, Valencia, CA, USA). The day before transfection, HMO6 cells were trypsinized, counted, and seeded in 35-mm culture plates at the appropriate density (4 × 10^5^ cells/mL). When cells were 80% confluent, the culture medium was changed to serum-free DMEM and pcDNA3.1(+). NEP and SRA were transfected into HMO6 cells using the PolyFect^®^ Transfection Reagent kit (Qiagen) and cultured for 24 h. G418 (500 μg/mL; Promega, Madison, WI, USA), an aminoglycoside antibiotic, was treated for four weeks until positive clones were selected [7]. The expressions of NEP and SRA mRNA were confirmed using primers with sequences forward: 5′-ATCAGCCTCTCGGTCCTTGT-3′; reverse: 5′-TGGAAGACAGCGCAAGACTC-3′ for NEP and forward: 5′-AGGAGATCGAGGTCCCACTG-3′; reverse: 5′-TGTTTCCACTCCCCTTTTCC-3′ for SRA.

#### 4.1.4. Immunocytochemical Analysis of Functional Proteins

Cultured F3, F3.ChAT, HMO6, HMO6.NEP and HMO6.SRA were fixed in 4% paraformaldehyde in phosphate-buffered saline (PBS) for 15 min, washed twice with PBS supplemented with 100 mM glycine for 5 min, and incubated with a permeabilization buffer consisting of 0.1% Triton X-100 (Sigma-Aldrich, St. Louis, MO, USA) in PBS for 30 min at room temperature. Blocking was performed with 1% bovine serum albumin (BSA, Sigma-Aldrich) for 30 min at room temperature. Then, ChAT (Invitrogen), SRA (1:100, Bioss, Woburn, MA, USA) or NEP (1:100, Invitrogen) primary antibody was added to 1% BSA in PBS with Tween 20 (PBS-T) and incubated for 2 h at room temperature. The cells were washed three times with PBS before fluorescein isothiocyanate (FITC)-conjugated or Alexa Fluor^®^ 555-conjugated anti-rabbit IgG 1:200 (Molecular Probes, Eugene, OR, USA) was added in 1% BSA for 1 h at room temperature. The cells were rinsed and counterstained with 4′,6-diamidino-2-phenylindole (DAPI) (Invitrogen) for 10 min, then observed after two PBS washes.

#### 4.1.5. Aβ Clearance by HMO6 Cells

HMO6, HMO6.NEP and HMO6.SRA cells were cultivated in 24-well plates at 1 × 10^5^ cells/well to analyze Aβ clearance. Cells were incubated containing Aβ (1 μg/mL) and 10% FBS, 100 U/mL penicillin and 100 μg/mL streptomycin (Invitrogen) for 1–3 h. The remaining Aβ concentration in the medium was analyzed using an Aβ ELISA kit (IBL, Naka Aza-Higashida, Fujioka, Gunma, Japan). The Aβ medium and Aβ standard supernatant were incubated in a pre-coated plate overnight at 4 °C. Each well of the plate was washed with washing buffer several times and then incubated with HRP-conjugated anti-human Aβ mouse IgG for 1 h at 4 °C. After washing, the cells were incubated in TMB solution for 30 min at room temperature in a dark room. The reaction was stopped by stop solution (1N H_2_SO_4_), and the yellow color was measured at 450 nm.

### 4.2. Establishment of AD Animal Model

#### 4.2.1. Animals

Six-week-old male ICR mice were procured from Daehan-Biolink (Eumseong, Republic of Korea). The animals were maintained at a constant temperature (23 ± 2 °C), relative humidity of 55 ± 10% and a 12-h light/dark cycle. The animals were fed with standard rodent chow and purified water ad libitum.

#### 4.2.2. Experimental Design for AD Modeling

To select a proper AD mouse model (n = 8/group), three neurotoxicants, Aβ42, AF64A and LPS, via different routes were assessed. To prepare fibrillary-aggregated Aβ42, lyophilized Aβ42 (Abcam, Cambridge, UK) was incubated at 37 °C for seven days at 100 μM in 10% DMSO solution in PBS (pH 7.4). AF64A (Sigma-Aldrich) was freshly prepared by treating with NaOH, followed by the adjustment to pH 7.4. LPS (Sigma-Aldrich) was dissolved in sterile saline [29,30,52].

The candidate neurotoxicants were injected (1-time point) once into mice ICV or intraperitoneally (IP). Only LPS(IP) group animals were injected seven times daily. For ICV injection, the skin on the parietal bones was incised, and a hole through the skull was made with a 3-mm dental drill. Then, the toxicants were infused with a 5-μL Hamilton syringe into the brain at the following stereotaxic coordinates from bregma: posterior 0.1 mm, lateral 2.5 mm, and ventral 3.5 mm. In addition, a one-mL insulin syringe was used for IP injection. After the challenge, short-term (3 days) and long-term (4 weeks) cognitive functions were assessed, and all animals were sacrificed for neuropathology.

#### 4.2.3. Cognitive Function Tests

For the evaluation of memory acquisition, the mice were subjected to a passive avoidance Shuttle box (ENV-010MD; Med Associates Inc, St. Albans, VT, USA) 3 days after AD induction. The trials were performed at 1-h intervals. The Shuttle box apparatus comprises two compartments with a lamp and a steel-grid floor for electric shock. In the trials, electric shock (1 mA for 2 s) was delivered when mice entered the dark compartment from the lightroom through a guillotine door. In addition, the retention time of stay in the light room from light-on was recorded. Finally, the end-point was set at 180 s, denoting full acquisition of memory.

For the evaluation of spatial memory, the mice were subjected to Morris’s water-maze test. The circular water bath, filled with water maintained at 22 ± 2 °C, was divided into four quadrants and a hidden escape platform (10 cm in diameter) was submerged in the center of 1 quadrant. The mice were trained to learn to find the hidden platform based on several cues external to the maze. In addition, the time spent to escape onto the platform was recorded.

#### 4.2.4. Western Blot Analysis of Brain Aβ

The mouse brain was collected after intracardial perfusion with cold saline at the end of learning and memory tests. The brain tissue was homogenized in RIPA buffer (Sigma-Aldrich) with protease inhibitors. After being centrifuged at 15,000 rpm for 15 min at 4 °C, total proteins were quantified with a BCA protein assay kit (Thermo Scientific, Waltham, MA, USA). Subsequently, tissue proteins were denatured by boiling for 5 min at 95 °C in 0.5 mol/L Tris-HCl buffer (pH 6.8) containing 10% sodium dodecyl sulfate (SDS) and 10% ammonium persulfate (APS), separated by electrophoresis on a 7.5% SDS-polyacrylamide gel (SDS-PAGE), and transferred onto a polyvinylidene difluoride membrane (PVDF) in 25 mmol/L Tris buffer containing 15% methanol, 1% SDS and 192 mmol/L glycine. After blocking for 2 h with 5% skim milk in TBS-T, the membrane was incubated with an antibody specific for Aβ (1:500, Millipore), overnight at 4 °C. After washing with TBS-T, the membrane was incubated with a secondary goat anti-rabbit IgG conjugated with horseradish peroxidase (1:1000, Cell Signaling Technology) for 2 h at room temperature. The membrane was then developed using an ECL solution (Thermo Scientific).

#### 4.2.5. Enzymatic Analysis of Brain ACh

The brain tissue was homogenized in 10 volumes of 0.1 mol/L sodium phosphate buffer and centrifuged at 13,500 rpm for 6 min at 4 °C to obtain supernatant. ACh concentration was measured using an Amplex^®^ Red Acetylcholine/Acetylcholinesterase Assay kit (Invitrogen). In the assay, ACh was hydrolyzed using AChE to release choline, and the choline was oxidized into betaine and hydrogen peroxide using a choline oxidase. The hydrogen peroxide interacted with Amplex Red (7-dihydroxyphenoxazine) in the presence of horseradish peroxidase, generating highly-fluorescent resorufin. The resultant fluorescence was measured using a fluorescence microplate reader (Bio-Tek Flx800; Bio-Tek Instruments, Winooski, VT, USA) with excitation in the 530–560 nm range and emission at ∼590 nm.

### 4.3. Efficacy Evaluation of NSCs and Microglial Cells Encoding Functional Genes

#### 4.3.1. Experimental Design for Efficacy Evaluation

Mice (n = 8/group) were anesthetized and positioned in a stereotaxic frame (Stoelting, Wood Dale, IL, USA). After the incision of the skin on the parietal bones, freshly-prepared AF64A solution (10 nmol/2 μL/mouse) was infused into the brain at the following stereotaxic coordinates from bregma: posterior 0.1 mm, lateral 2.5 mm and ventral 3.5 mm. Five days later, the mice were transplanted with F3, F3.ChAT, HMO6, HMO6.NEP and HMO6.SRA cells alone or in combination with F3.ChAT + HMO6.NEP or F3.ChAT + HMO6.SRA cells in a mixture (4 μL/mouse) via left lateral ICV injection at the following coordinate: posterior 1.0 mm, lateral 2.0 mm and ventral 3.0 mm. A schematic presentation of the experiments is shown in Appendix A.

#### 4.3.2. Cognitive Function Tests

For the evaluation of cognitive function, including memory acquisition and spatial memory, passive avoidance and Morris’s water-maze performances were assessed as described above four weeks after cell transplantation.

#### 4.3.3. Immunohistochemical Analysis of hMito and Functional Proteins

The mouse brains were perfusion-fixed with 4% paraformaldehyde solution and post-fixed for 48 h, followed by cryoprotection in 30% sucrose for 72 h. Coronal cryosections in 30-μm thickness were prepared and processed for double immunostaining of human mitochondria (hMito) and/or ChAT, NEP or SRA using antibodies specific for hMito (1:100, Millipore), ChAT (1:100, Invitrogen), NEP (1:100, Invitrogen) and SRA (1:100, Bioss). Brain sections were incubated with primary antibodies overnight at 4 °C and with secondary antibodies conjugated with Alexa Fluor 488 or 594 (1:1000, Molecular Probes) for 1 h at room temperature.

#### 4.3.4. ELISA and Enzymatic Analyses of Brain Aβ and ACh

Aβ concentration in the brain homogenates was analyzed using an Aβ ELISA kit (IBL) as described above. Brain ACh concentration was analyzed using an Amplex^®^ Red Acetylcholine/Acetylcholinesterase Assay kit (Invitrogen) as described above.

#### 4.3.5. Western Blot Analysis of Brain Aβ and GFAP

From whole brain homogenate, proteins were separated by electrophoresis and transferred onto a PVDF membrane as described above. The membrane was incubated with antibodies specific for Aβ (1:500, Millipore) or GFAP (1:1000, Millipore) overnight at 4 °C. The membrane was incubated with a secondary goat anti-rabbit IgG conjugated with horseradish peroxidase and then developed using an ECL solution.

### 4.4. Statistical Analysis

The results are presented as mean ± standard deviation. The significance of differences among results was analyzed by one-way analysis of variance, followed by post-hoc Tukey’s multiple-comparison tests. *p*-values < 0.05 were considered statistically significant.

## Figures and Tables

**Figure 1 ijms-24-09561-f001:**
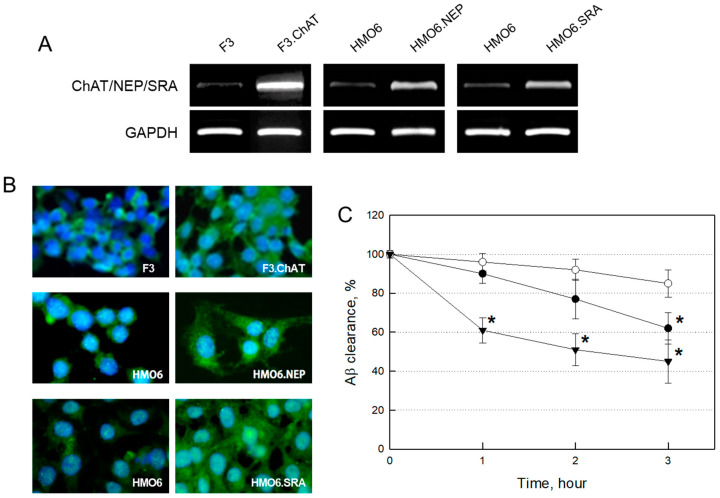
Expression of mRNAs and production of functional proteins in human neural stem cells and microglial cells, and amyloid-β (Aβ) clearance by microglial cells. (**A**) RT-PCR analysis of ChAT, NEP and SRA mRNAs. (**B**) Immunocytochemical staining on ChAT, NEP and SRA (green). (**C**) Clearance of Aβ peptides by HMO6 (○), HMO6.NEP (●) and HMO6.SRA (▼). * Significantly different from HMO6 cells at each time point (*p* < 0.05).

**Figure 2 ijms-24-09561-f002:**
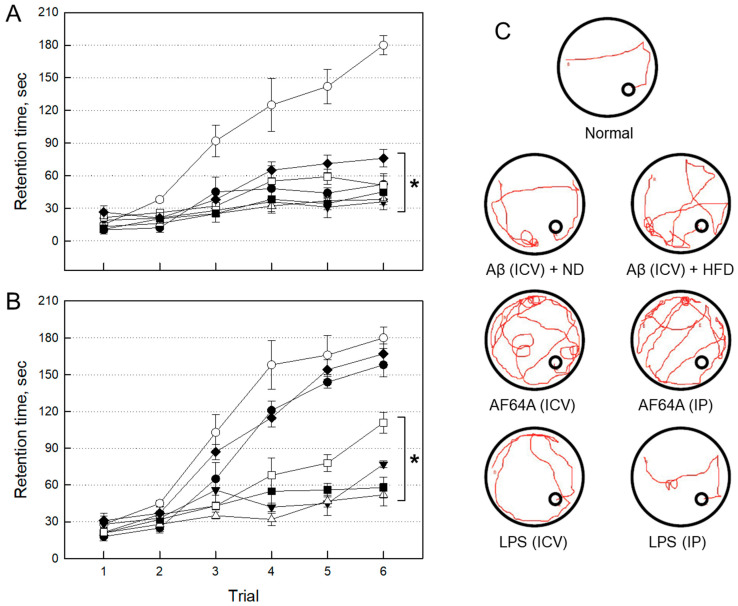
Impairment of learning and memory functions three days (**A**) and four weeks (**B**) post-injection in passive avoidance test and tracking of Morris water-maze swimming performance (**C**). ○: Normal, ●: Aβ (100 μM, 5 μL/mouse, normal diet) ICV injection, ▼: Aβ (100 μM, 5 μL/mouse, high-fat diet) ICV injection, △: AF64A (10 nM, 2 μL/mouse) ICV injection, ■: AF64A (23.7 μmol/kg) IP injection, □: LPS (1 μg/5 μL/mouse) ICV injection, ◆: LPS (250 μg/kg, for seven days) IP injection, ND: normal diet, HFD: high-fat diet, ICV: intracerebroventricular, IP: intraperitoneal, LPS: lipopolysaccharide. * Significantly different from normal (*p* < 0.05).

**Figure 3 ijms-24-09561-f003:**
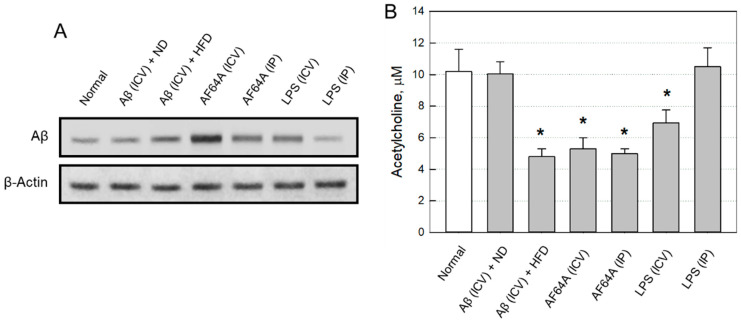
Amyloid-β (Aβ) accumulation (**A**) and acetylcholine depletion (**B**) in the mouse brains four weeks post-injection of toxicants. ND: normal diet, HFD: high-fat diet, ICV: intracerebroventricular, IP: intraperitoneal, LPS: lipopolysaccharide. * Significantly different from normal (*p* < 0.05).

**Figure 4 ijms-24-09561-f004:**
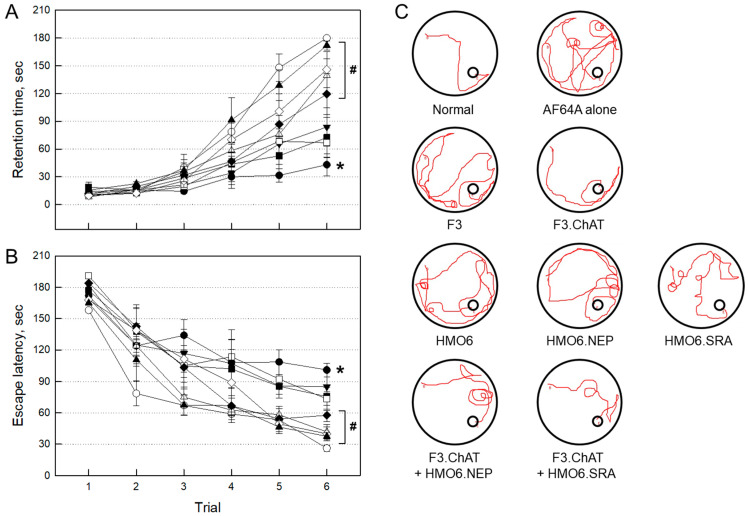
Recovery of learning and memory functions four weeks post-injection in passive avoidance (**A**) Morris water-maze (**B**) studies, and tracking of water-maze swimming performance (**C**). ○: Normal, ●: AF64A alone, ▼: F3, △: F3.ChAT, ■: HMO6, □: HMO6.NEP, ◆: HMO6.SRA, ◇: F3.ChAT + HMO6.NEP, ▲: F3.ChAT + HMO6.SRA. * Significantly different from normal (*p* < 0.05). ^#^ Significantly different from AF64A alone (*p* < 0.05).

**Figure 5 ijms-24-09561-f005:**
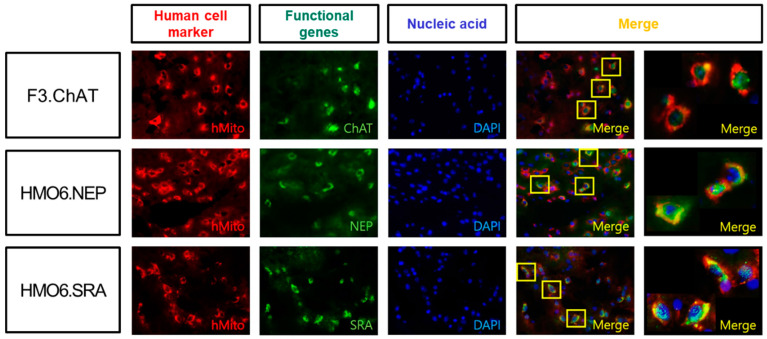
Immunohistochemical identification of human neural stem cells (F3) and microglial cells (HMO6) and functional proteins (ChAT, NEP and SRA) via double immunostaining for human mitochondria (hMito) and ChAT, NEP or SRA.

**Figure 6 ijms-24-09561-f006:**
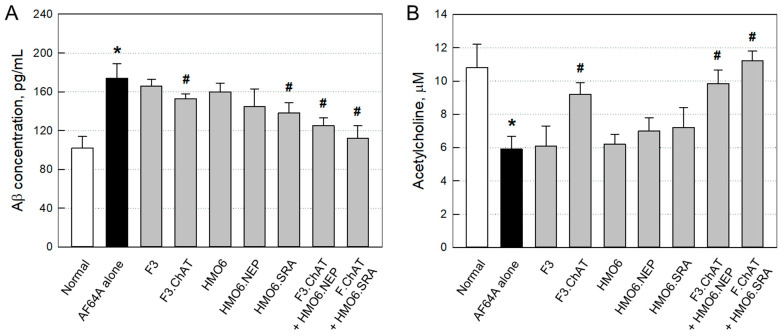
Recovery of brain amyloid-β (Aβ, (**A**)) and acetylcholine (**B**) levels in mice challenged with AF64A 4 weeks after cell transplantation. * Significantly different from normal (*p* < 0.05). ^#^ Significantly different from AF64A alone (*p* < 0.05).

**Figure 7 ijms-24-09561-f007:**
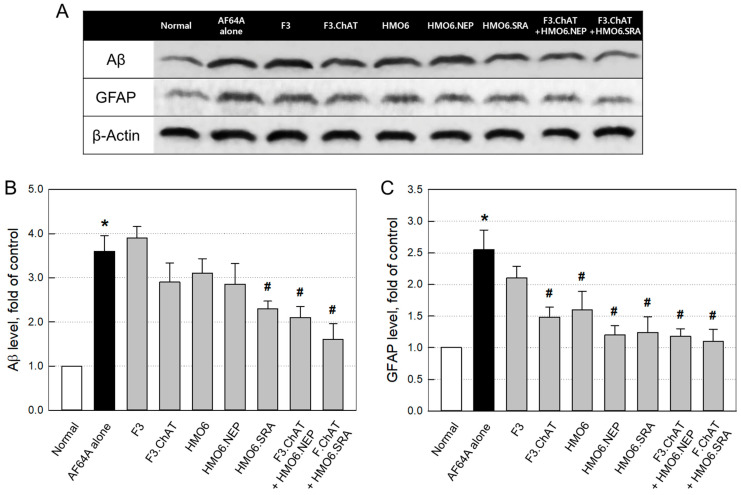
Representative western blot bands (**A**) and the changes in the brain amyloid-β (Aβ) (**B**) and glial fibrillary acidic protein (GFAP) (**C**) levels in mice challenged with AF64A 4 weeks after cell transplantation. * Significantly different from normal (*p* < 0.05). ^#^ Significantly different from AF64A alone (*p* < 0.05).

## Data Availability

Data are contained within the article or Appendix A.

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
