# Peer review of "Effectiveness of Combinational Treatments for Alzheimer’s Disease with Human Neural Stem Cells and Microglial Cells Over-Expressing Functional Genes"

_ijms, 2023, doi:10.3390/ijms24119561_

Round 1
Reviewer 1 Report
Overview
The “Effectiveness of Combinational Treatments of Alzheimer’s Disease with Human Neural Stem Cells and Microglial Cells Over expressing Functional Genes” demonstrates Alzheimer’s Disease (AD) statement and treatment of stem cells and genes expressing. The study used F3.ChAT human neural stem cells (NSCs), HMO6.NEP human microglial cells and HMO6.SRA cells to culture and encoding genes to measure effects in animal model.
1. Purpose of the article: study of the effectiveness of combined stem cell and myoglia therapy in the treatment of Alzheimer's disease
2. research materials: experimental mice were used as the material. Consists of a control group and several study groups. Stem cells and myoglia were transplanted into each group of mice after 3 weeks, the decrease in beta amyloid and their effect on memory in Alzheimer's disease was monitored
3. Conclusion: Combination therapy in Alzheimer's disease was effective in experimental mice memory improved through beta amyloid cleavage.
The subject of the article is not new as it discusses the potential of cell-based therapies for Alzheimer's disease, which has been a topic of research and discussion for many years. However, it discusses about expressing Functional Genes of AD treatments through stem cells which are wide in future.
Comments
1. There are some concern about significancy in some graphs
2. All graphs and statistical data must be checked again.
3. Line 19, not use (,) before the word and.
4. Line 24, what does it mean alone or combination? Does it follow 2 different ways for doing the research?
5. Please correct the keywords through the MESH. It seems so much long words.
6. ChAT cells have been written in complete form in previous lines. But, F3.ChAT cells have no description or full name mentioned in line 66.
7. Line 67, what is NSC line HB1.F3? however NSC were mentioned in full name in previous lines, may it need to describe more about line HB1.F3.
8. Lines 83 to 85, it is introduction, why is seemed that you are reporting the results?
9. Lines 83,87,88 talk about HMO6 cells but without explaining them. Only in line 26 in abstract HMO6 cells were explained. So, if a person wants to read introduction first, how can he know?
10. Lines 103 and 104, why it has been used or and and both? Please make declaration and clear it if each single cell were study alone or all kind of cells were studied together as a group. If it is alone please say and clear and if it is combination of them please make it clear. Even if both were examined please mention that clearly and do not make doubts.
11. Why Results is before Methods? And Methods is at the end? Please follow the routine layout.
12. Line 126, please make sense between IP and IV and make it clear.
13. For Methods part it was better if there were photos of steps.
14. Line 237, not use (,) before the word and.
16. Line 330, please write how fetal bovine serum (FBS) were collected.
17. Line 335 please write how 15-week gestational human fetal brain were collected.
18. Line 385, I suppose that the word well needs change to be cell.
19. Part animals please write the moral code.
20. Lines 403 & 404 please make it clear what kind of way did done to injection. Finally ICV or IP or IV? If both ways were used please not use or.
21. Please notice how many mouses got injection.
22. Line 404, how many times ICV group were injected? It doesn’t mentioned.
23. Line 424, proteins of what?
24. Line 422, please mention how mouse brain was collected.
25. Line 454, make it sure that cells were transplanted together or alone.
26. One of my colleagues said that there is no bioethical protocol in the text of the article, but I consider the use of artificially removed cells or products obtained by culturing them to be carried out by specialized institutions that have legislative documents (accreditations, certificates).
27. The statistics used mean (М) and standard error(m).
28. During the visual analysis presented in the diagrams, errors in the statistical aspect were revealed. Since I am not familiar with the discussion section of this article, therefore, I am not able to give a complete negative characterization of these figures. I think the authors in the text of the article gave an explanation for these unaccounted for.
29. Apparently, this article is under review, as the editor of the journal, after correcting the statistical data, I would give permission for publication, because the materials of the article can be used to treat patients with Adzheimer's disease in the future.
30. Stem cell therapy is currently considered a relatively new area of medicine, where it is used in the treatment of Alzheimer's disease, Parkinson's disease, autism, etc. These are diseases for which the cause is still unknown. And the use of this type of therapy is aimed at improving self-regulatory mechanisms using transplantation of similar cells.
Author Response
Please see the attach answers. Thank you!

Reviewer 2 Report
Add more information about cell localization and migration in the nervous system of mice. Add sections of mouse brains where cell localization is observed.
Some of these cells are immortalized with viruses. Discuss the possibility that they generate tumors.
How was the treatment with Neuronal Stem Cells conducted?
How were they injected into the brains? How do you think injecting these cells into Alzheimer's patients brains?
Can these cells differentiate?
Do you have rejection data on such cells? Since they are human cells implanted in mice.
Do you think this is symptomatic or asymptomatic therapy? It is quite elaborate to perform. Considering the possibility of rejection by not only the mouse brain but also the human brain.
I was not able to find the supplementary materials.
Author Response
Please see the attached answers. Thank you!

Round 2
Reviewer 1 Report
In the first review of the article, it was pointed out that the graphs and the display of significance were done according to the statistical tests, and considering that it is mentioned in the statistical analysis section that the error bars are standard errors, therfore the tests are not correct. Therefore, there is a great concern that the statistical analysis needs to be done again and it is not acceptable with the current results of this paper. Unless the deviations are standard deviations and not standard errors. Please pay attention to these examples:
Figure 3 in the third hour related to the HMO6.NEP group, this significance is true if the standard deviation is correct. But it has been found that this is a standard error.
Significances are not shown in Figure 2, part A.
The significance in Figure 6, part A, in the case of F3.ChAT is not correct if the bar is the standard error.
and other similar cases. Therefore, it is highly recommended to check the statistical analysis.
Author Response
Thank you for your valuable advice!
Please see the attached answers.

Reviewer 2 Report
The authors added the information that I had requested. Therefore the manuscript can be published. It is an excellent basic research study. Still, some concern remains about its applicability to humans.
Author Response

(The authors gave the same response as above.)

Round 3
Reviewer 1 Report
I dont have more comment.
Author Response
Please find the attached Editor's Notes.
